# Staying under the Radar? Immigration Effects on Overdose Deaths and the Impact of Sanctuary Jurisdictions

**Kelly Pierce** [1,*] , **Diana Sun** [2] **and Ben Feldmeyer** [1]

1   School of Criminal Justice, University of Cincinnati, Cincinnati, OH 45241, USA
2   School of Criminology and Criminal Justice, Florida Atlantic University, Boca Raton, FL 33431, USA
*   Correspondence: piercek2@mail.uc.edu

**Abstract:** Growing political and public rhetoric claim that immigration has contributed to drug crime and the overdose crisis of the 21st century. However, research to date has given little attention to immigration–overdose relationships, and almost no work has examined the ways that the sanctuary status of locales influences these connections. The current study draws on the immigrant revitalization perspective and Brayne's (2014) systems avoidance theory to examine the connections between immigrant concentration, sanctuary status, and overdose mortality across MSAs for the 2015 period, overall and across races/ethnicities. The analysis uses data on overdose deaths drawn from the CDC's Restricted Access Multiple Cause of Death Mortality files, combined with data on characteristics of MSAs drawn from the U.S. Census and other macro-level data sources. Findings reveal that the percent Latinx foreign-born is related to lower levels of overdose deaths overall and for White and Black populations but higher levels of Latinx overdose mortality. Contrary to expectations, sanctuary status has little effect on overdose deaths across most groups, and it does not significantly condition immigration–overdose relationships.

**Keywords:** communities and crime; sanctuary; drugs; immigration; macro-level; overdose





## 1. Introduction

Beginning around the turn of the 21st century, the United States entered an overdose crisis that has spanned more than two decades and claimed the lives of nearly 1 million Americans [1,2]. U.S. deaths from drug overdoses increased by roughly 15% from 93,655 in 2020 to 107,622 deaths in 2021 [3]. Prior to that, drug overdose deaths soared by 30% from 2019 to 2020 [3]. As overdose deaths continue to mount, the CDC has recognized the overdose epidemic as one of the deadliest U.S. drug and health crises on record. At the same time, the United States has also seen high levels of immigration, leading some to suggest that it may be contributing to the drug crisis. Although research has consistently shown that immigration is not linked to higher levels of crime and violence [4–8], there has been growing political rhetoric suggesting that immigration is partly responsible for escalating drug and overdose problems. For example, during former President Donald Trump's state of emergency address in 2019, he attributed the rise of drugs in the United States to undocumented immigrants, stating, "We're talking about an invasion of our country with drugs, with human traffickers, with all types of criminals and gangs" [9].

Despite widespread political rhetoric on the topic, research to date has given surprisingly little attention to the macro-level relationships between immigration and drug use. In addition, even less work has examined immigration–overdose relationships with some notable exceptions [10,11], and no studies to date have examined the ways that "sanctuary cities" or "sanctuary status" shape overdose mortality and its connections with immigration. As we describe in the following sections, there are particularly important reasons to examine sanctuary status in studies of immigration and overdose and reasons to expect that it could shape drug-related deaths.

Research suggests that immigrant communities and especially Latinx immigrant populations may be hesitant to use official sources of help for drug treatment due to fears of deportation and the criminalization of both addiction and immigration [12,13]. In particular, Brayne's [14] description of "systems avoidance" theory explains how these fears keep immigrant communities from contacting record-keeping agencies. Although they vary widely by location, sanctuary policies generally refer to placing limits on cooperation between local agencies and federal immigration enforcement, and they have been touted as a tool to help reduce these fears and increase connections between immigrant communities and agencies that could offer aid, such as law enforcement, hospitals, and emergency medical services [15–17]. With respect to overdose, sanctuary status may alleviate fears about these institutions and encourage help seeking for drug treatment that is key to reducing fatal overdoses. However, almost no research to date has examined sanctuary policies and their impact on overdose deaths.

In light of the issues highlighted above, the current study is one of the few analyses to date examining immigration and overdose relationships. We also extend this work by drawing on "systems avoidance" theory (which has received little attention in immigration-crime studies) to explore how sanctuary status shapes overdose mortality and its connections with immigration, overall and across White, Black, and Latinx groups. To do so, we examine relationships between immigrant concentration, sanctuary city status, and overdose deaths across MSAs (Metropolitan Statistical Areas) for the year 2015 using data drawn from the CDC's Restricted Access Multiple Cause of Death Mortality (MCOD) data alongside information on sanctuary city status, immigration, and other characteristics of MSAs drawn from U.S. Census data and records on sanctuary city status.

## 2. Literature Review

### 2.1. Immigration, Crime, and Drug Use

The United States has a long history of fear about immigration–crime connections, and these fears have only been exacerbated with increasingly divisive rhetoric on the topic. From journals decrying the "menace" of Mexican immigrants destabilizing the country via marijuana [18] to former President Trump's more recent assertions that immigrants are responsible for bringing in drugs, crime, and rape [9,19], immigrants have often been a target of blame for social ills. These concerns have been used as a justification for a variety of punitive immigration policies in the name of protecting citizens [20]. Despite these fears, research shows no consistent evidence linking immigration to higher crime rates [5]. In fact, some of the most drastic crime drops seen around the turn of the 21st century occurred in areas with high concentrations of immigrants [6,21,22], but see [23,24]. In sum, there is now a large body of research showing that, in contrast to public fears and political rhetoric, immigration is not associated with higher levels of crime and violence and is often linked to lower crime rates. However, there is surprisingly little research examining whether and how immigration is connected to serious drug use and to rates of overdose deaths.

This oversight is surprising for several reasons. First, both immigration and overdose deaths have escalated throughout the first two decades of the 21st century. In addition, there has been no shortage of political rhetoric suggesting that immigration is responsible for drug crime and serious drug problems such as overdose deaths throughout U.S. communities [25]. (For example, former President Trump claimed, "We have tremendous amounts of drugs flowing into our country. Much of it coming from the southern border, when you look and when you listen to politicians, particularly certain Democrats, they say 'It all comes through the ports of entry.' Wrong, it's wrong. It's just a lie. It's all a lie" [9]. Moreover, Gallup polls suggest that almost half of the U.S. public believes that immigration has heightened the crime problem in the country [26]. Yet, some research suggests otherwise.

Many individual-level studies have shown that immigrants and Latinx immigrants in particular have lower levels of drug and alcohol use than comparable U.S. born individuals [27–30]. In a comparison of Mexican-Americans born in the United States versus those

born in Mexico, Borges and colleagues [27] found that the native-born members of their sample were more likely to suffer from drug dependence and misuse than those born in Mexico. There are undoubtedly drug cartels that operate across borders and bring in illicit drugs [31], but there is little evidence that immigrants are more broadly responsible for increased substance use. Instead, researchers have typically found lower levels of substance abuse and drug-related arrests in areas with large immigrant communities [11,32,33] and lower levels of drug crime and drug use among immigrants themselves [27,34–36].

However, there are remarkably few macro-level studies examining immigration and drug use, and even fewer that specifically focus on the 21st century overdose crisis. In fact, a review of the literature revealed only two studies to date that have specifically examined connections between immigration and overdose death rates. In one study, Feldmeyer and colleagues [37] examined how immigration flows were linked to county-level rates of both overdose and homicide from 2000 to 2015. They found that increases in immigration over time were associated with significantly fewer overdose deaths overall and for multiple different substance types. Although this analysis was focused on general immigrant populations, supplemental analyses looking at Latinx immigration levels showed similar findings. In a second study, Light and colleagues [11] examined how changes in state-level undocumented immigrant populations were associated with substance use, DUI, and overdose trends from 1990 to 2014. They found no signs of increased substance use or overdose deaths in areas with large undocumented populations. Instead, they reported a negative or null relationship between undocumented immigration levels and all measured outcomes.

Taken together, the few studies examining this topic suggest that immigration does not increase overdose deaths, and in fact, may help lower overdose mortality. However, research on the topic remains scarce, and as we highlight in the following section, there are theoretical arguments suggesting that immigration could either increase or decrease rates of drug use and overdose deaths. Furthermore, there are reasons to expect that sanctuary city status may have important effects on these relationships, which we discuss in further detail below.

*2.2. Reasons Immigration Could Increase Overdose Deaths*

Although immigration does not appear to increase crime, there are several theoretical arguments that suggest that immigration flows could be linked to higher rates of crime and serious drug use. These perspectives have been discussed extensively in prior immigration–crime literature; thus, we refer the reader to those works for more detailed treatments of these theories [5,37–39]. To briefly summarize, however, these arguments suggest that high levels of immigration could contribute to crime and related behaviors by increasing community patterns of strain and social disorganization. Immigrant populations tend to be disadvantaged and have greater educational and financial deprivation than many native-born populations [4,38,40,41]. They may also face a variety of non-financial strains and challenges associated with settlement in a new environment (e.g., language barriers, navigating immigration processes, finding work, housing, transportation, and childcare [42–45]. In addition, the influx of disadvantaged groups with different languages and norms could inhibit community organization and informal control over crime according to traditional social disorganization arguments [46]. These perspectives have typically been applied to studies of violence, but they also suggest potential immigration–drug relationships.

Although it has received limited attention in immigration-crime research, *systems avoidance* theory also offers reasons why immigration could be linked to greater overdose deaths. Systems avoidance theory, described by Brayne [14], refers to the practice of avoiding contact with systems that maintain formal records out of fear of surveillance and apprehension. With increasing levels of surveillance among society's institutions, Brayne [14] argues that people who may be at risk for apprehension adapt their behavior to minimize contact with record keeping institutions, such as law enforcement, hospitals, banks, schools, and government offices.

Brayne's [14] original application of systems avoidance focused on those with criminal justice records and found that individuals with prior arrests, convictions, or incarceration experience were significantly less likely to engage with banks, schools, and hospitals. Informal institutions and formal institutions that do not engage in record keeping, such as religious institutions or informal volunteer agencies, were not impacted by prior criminal justice involvement. These findings suggest deliberate actions on the part of the individuals to avoid detection and potential apprehension. Although Brayne [14] specifically focused on systems avoidance for those with a criminal record, this concept may extend to other groups that would benefit from avoiding detection and recordkeeping institutions—including immigrants.

Related to this perspective, there is a growing line of research showing a reluctance among immigrant communities to cooperate with criminal justice agencies [8,47]. For example, studies show that immigrant populations may be hesitant to report crimes and victimizations, serve as witnesses for police investigations, and cooperate with court proceedings [8,48–51]. Research suggests that fear of deportation and disruption to their lives (either for themselves or family and friends) may be driving much of this criminal justice avoidance [8,47–51]. However, these studies have focused largely on criminal justice agencies (see [51] for exception) and have given less attention to other record-keeping institutions that could impact crime, drug use, and overdose.

Systems avoidance theory suggests that immigrant communities may be hesitant to interact with hospitals and medical providers, which could have key consequences for drug overdose risks. When considering the dual criminalization of immigration [52] and substance use through the War on Drugs era [53], contacting formal institutions for addiction treatment may seem too risky, especially among undocumented populations. Systems avoidance effects may also be more pronounced among Latinx populations, as the risk of deportation in the United States is highest among this group [54,55]. Although Latinx overdose rates are generally lower than for other groups, these numbers are increasing [56–58] and are thus important for examination when assessing immigration–overdoses relationships.

### 2.3. Reasons Immigration Could Decrease Overdose Deaths

In contrast to the perspectives described above, there are also plausible reasons to expect that immigration may reduce (or have neutral effects on) community levels of serious drug use and overdose deaths. According to the immigrant revitalization perspective, immigration has protective effects that help insulate communities from crime and related social problems. The immigrant revitalization thesis suggests that immigrants bring benefits to their communities, such as job growth and strong social capital networks [5,6,59,60]. Despite facing many structural disadvantages (e.g., lower incomes, education levels), this perspective suggests that immigrant populations are able to rebuff criminogenic influences due to their strong cultural proclivities and social networks.

For instance, immigrants may move into areas with pre-existing kinship ties and strong community values based on a common ancestry [5,61–65]. As such, they are likely to have strong informal social control on behaviors such as crime. Furthermore, immigrants may infuse local economies, assist with job creation, and decrease levels of vacant housing—overall enriching the areas they move into [6,63]. In support of these arguments, researchers have repeatedly found that immigration has either no effect or is linked to lower crime [5,37,63–70], including areas with large undocumented populations [71].

To date, this perspective has mostly been applied to studies of immigration and violence. However, there are clear reasons to expect that the protective, revitalizing effects of immigration may also help reduce drug use and overdose death rates. The few existing studies that have examined immigration–overdose relationships provide support for revitalization arguments, showing that, if anything, immigration has been associated with lower levels of drug overdose deaths across multiple different substance types [11,37]. However, additional work is needed to understand more fully the effects of immigration on overdose mortality, especially in sanctuary jurisdictions, which is a focus of the current study.

### 2.4. Sanctuary Jurisdictions

Sanctuary cities and sanctuary status have received little attention in immigration-crime research, and even less attention in studies of immigration and drug use. Yet, they provide an important context that could shape these relationships. Across the United States, an increasing number of urban areas have designated themselves as "sanctuaries" [16]. Though political rhetoric has brought renewed attention to sanctuary jurisdictions [72,73], the initial sanctuary movement in the U.S. occurred in the 1980s, and was largely driven by religious institutions offering aid and protection from immigration authorities to undocumented refugees fleeing El Salvador and Guatemala [74]. In 2007, renewed interest in the sanctuary movement occurred [74] as the criminalization of migrants increased and the use of incarceration without due process expanded [75], which was met with heightened concerns about protecting immigrants. This concern continued throughout the 2000s and 2010s as the country ramped up aggressive deportation and removal of immigrants throughout the Obama administration. More recently, there was a large spike in enacted sanctuary policies in response to Trump's declared anti-immigration stance [76]. At the time of this this writing, there is no standard definition for "sanctuary status" beyond some level of local non-cooperation with federal immigration enforcement [16]. However, this may take a variety of forms, including formal statements of protections or simply informal practices of non-cooperation with immigration enforcement.

Given the somewhat subjective nature of what qualifies as sanctuary status, there are no conclusive numbers about the prevalence of sanctuary jurisdictions. Nevertheless, there has been a clear expansion in sanctuary status over the last 20 years—estimates suggest that at least 328 sanctuary policies were enacted between 2009 and 2017 [77]. Even among formal policies, there is significant variation in the details [78]. Policies may generally state that local police are not responsible for and should not participate in immigration enforcement, or they may include specific directives informing police of what actions are expected regarding specific enforcement action (or inaction) and the level of cooperation with ICE required [79]. For example, these directives may include instructing police not to inquire about a suspect's immigration status, or it may direct sheriffs to refrain from accepting ICE detainers, which refer to requests to hold suspected undocumented immigrants for 48 hours beyond release to provide additional time for deportation actions to be taken [16]. It should be noted that even in some jurisdictions such as Los Angeles that are considered to have the most protective and pro-immigrant sanctuary policies, there are limits to the protections offered. For example, Los Angeles still require officers to cooperate with ICE in the event of serious criminal behavior [80].

A major justification for adopting sanctuary status is that these policies are thought to encourage cooperation between law enforcement and immigrant communities. The goal is to limit the "chilling effect," fears of deportation, and broader systems avoidance that can occur if local law enforcement becomes associated closely with immigration enforcement [79]. Adopting sanctuary status is thought to increase reporting, cooperation, and help seeking between immigrant communities and law enforcement, but it may extend to other service providers including hospitals and medical care.

Few studies to date have specially investigated the role of sanctuary status in shaping reporting and cooperation levels, but among those that have, findings indicate that sanctuary status can improve reporting behavior [49,78]. With respect to overdoses and serious drug use, sanctuary cities may be important for reducing fears associated with reporting overdoses, seeking treatment, and contacting service providers (including hospitals, doctors, law enforcement, and emergency medical services), which in turn could help reduce or prevent overdose deaths. Likewise, the immigrant revitalization perspective would also suggest that sanctuary cities could be beneficial for reducing overdose deaths and serious drug use. Prior research has shown that the protective effects of immigration are bolstered in locations where the reception to immigrant settlement is inviting and less restrictive [22]. Although research has given little attention to sanctuary status effects on drug use, studies have found null or negative relationships between sanctuary status and

overall crime rates [17,72,81–84]. Taken together, the theories described here related to immigration and overdose both suggest that sanctuary cities could help reduce overdose deaths and further enhance the protective effects of immigration on serious drug use.

*2.5. The Current Study: Systems Avoidance, Immigrant Revitalization, and Sanctuary Status*

In light of the arguments described in the preceding sections [50], the goal of the current study is to assess the relationships between immigration, sanctuary status, and overdose death rates among MSAs in 2015. As reviewed above, theory provides alternative theoretical expectations about the ways that immigration is associated with drug overdose mortality. On the one hand, systems avoidance theory [14] (alongside strain and social disorganization perspectives) suggests that communities with larger concentrations of immigrants may see higher overdose rates due to fears associated with contacting local service providers—including law enforcement, hospitals, doctors, and emergency medical services. Based on these perspectives, we would expect the following:

**Hypothesis 1a.** *MSAs with larger immigrant populations will have higher levels of overdose deaths compared to other areas, especially for Latinx populations (who account for greater shares of immigrant populations).*

On the other hand, the immigrant revitalization perspective offers an alternative hypothesis. This perspective suggests that immigrant concentration has protective effects that help offset crime and related social problems, possibly including serious drug use and overdose deaths [37]. Based on this thesis, we present the following, competing hypothesis:

**Hypothesis 1b.** *MSAs with larger immigrant populations will have similar or fewer overdose deaths compared to other areas, especially for Latinx populations (who account for greater shares of immigrant populations).*

Furthermore, a key contribution of this study is that we incorporate sanctuary status to assess whether and how it is related to overdose mortality. As reviewed in the theoretical discussion above, both systems avoidance theory and the immigrant revitalization perspective suggest that sanctuary cities may foster greater help seeking and cooperation with key service providers. In addition, by providing a favorable reception for immigrant populations, sanctuary cities may help further reinforce many of the protective benefits that immigration has been shown to provide in reducing crime and related social problems. Based on these arguments, we predict the following:

**Hypothesis 2.** *MSAs with sanctuary policies in place will have fewer overdose deaths than those without sanctuary policies, net of other factors.*

**Hypothesis 3.** *The protective effects of immigration on overdoses deaths will be stronger in MSAs with sanctuary policies (i.e., an interaction between immigration levels and sanctuary status).*

## 3. Materials and Methods

To analyze the relationships between overdose deaths, Latinx immigrant populations, and sanctuary cities, data for the current project are drawn from several sources for the year 2015 at the MSA-level, which is our unit of analysis. MSAs are defined as geographic areas consisting of a county or group of counties that contain at least one urbanized area with a sufficiently large population (50,000 residents or more) and the surrounding counties with economic ties [85]. Information on drug overdoses is drawn from the CDC MCOD Mortality data [86]. The CDC MCOD Mortality data provide information on all death records in the United States from 2014 to 2016 and include the underlying cause of death (i.e., overdose), other contributing causes of death such as the specific substance type involved based on toxicology reports, and the location for each death.

The restricted-access MDOC data are particularly well suited for the analysis for several reasons. First, the data provide some of the most comprehensive records available on drug overdoses covering the United States. The data also provide death records that can be aggregated to MSAs, overall and across race/ethnicity. Second, the restricted-access MCOD data provide death records for communities with less than 10 overdoses in a given year, unlike the CDC public-access mortality death files, which only report places with at least 10 overdoses [37,86]. Information on the social-economic, legal, and other ecological characteristics of MSAs was drawn from the following sources to create our independent variables: (1) U.S. Census data (decennial and American Community Survey [ACS], 5-year estimates 2013–2017), (2) CDC Opioid Prescription data, and (3) UCR data. Information on sanctuary status of MSAs was drawn from a variety of sources that detail sanctuary policies discussed in detail below.

As noted above, our unit of analysis is MSAs, which are a common study unit used in prior research on immigration and crime [49,70,87,88] as well as in studies examining reporting behavior in sanctuary cities specifically [78]. MSAs cover large cities and counties, which is important given that sanctuary policies may be enacted at either level [16]. Given the rare nature of overdose death, MSAs are also large enough to allow meaningful statistical analysis including comparisons between sanctuary and non-sanctuary areas.

### 3.1. Dependent Variables

The dependent variable in our study is *overdose death counts*, overall and across race/ethnicity (Latinx, Black, and White). Overdose death figures were calculated using three-year averages around 2015 (2014–2016) to establish stability and minimize the impact of a one-year fluctuation in overdose deaths.[1] Overdose deaths by race/ethnicity are included to assess whether effects of immigration and sanctuary status are limited to those groups that have larger concentrations of immigrants (i.e., Latinx) or whether they have broader effects overall and across race/ethnicity. Notably, these groups reflect three of the largest racial/ethnic populations in the United States. However, expanding this analysis to include other racial/ethnic groups (e.g., Asian and Native American) is an important area for future inquiry.

### 3.2. Independent Variables

The first independent variable of interest, *percent Latinx foreign-born*, is drawn from the U.S. Census data to estimate the effects of immigrant populations (specifically of Latinx ethnicity) on overdose deaths overall and across race/ethnicity. In supplemental models, we replicated all analyses using percent foreign-born (rather than Latinx foreign-born), which produced similar results.[2]

The second independent variable of interest, *sanctuary status*, reflects MSAs with specific formal resolutions in place as of 2015, following the practices of prior research [72,78,83].[3] These formal resolutions must cover the majority of an MSA area to be classified as having sanctuary status. The variable was coded dichotomously (1 = sanctuary status, 0 = non-sanctuary) after excluding MSAs in which sanctuary status could not be determined or covered only a portion (less than 50%) of the MSA population.[4]

A variety of sources were used to locate this information, including previous studies of sanctuary areas at the MSA level [49,78], data from the International Immigration Law Center public data, FAIR US, maps from Center for Immigration Studies, and Ohio Jobs and Justice PAC.[5] In the case of conflicting information on sanctuary status across sources, the lead author independently verified the status of sanctuary policies for an MSA.

### 3.3. Control Variables

Control variables that may be related to immigration, overdose, and sanctuary status include *total population*, *percent uninsured*, *opioid prescription rate*, *residential mobility*, *police presence* and *disadvantage indices* (overall and for each race/ethnic group). Variables were primarily gathered from 2015 U.S. Census data (American Communities Survey, 2013–2017).

*Total population* is an important control given the count nature of the dependent variable, but it is also necessary in analyzing the sanctuary status, as sanctuary cities tend to have larger than average populations [72]. *Percent uninsured and opioid prescriptions* (rate per 100,000 residents) are included to ensure that any differences observed in statistical analysis are not due to geographic or racial/ethnic differences in access to healthcare or opioids. *Police presence* was calculated as a rate of total police per 100,000 residents [91,92]. *Residential mobility* represents the percentage of an MSAs population that lived in a different residence in the preceding year. The *disadvantage indices* were created to account for socio-economic conditions of MSAs and to address potential multicollinearity between measures of social and economic conditions (percent without high-school diploma, percent female-headed households, percent living in poverty, and percent unemployed).[6] These items returned acceptable alpha values overall and for all race/ethnic groups (near or above 0.70) and loaded onto one factor for each group.

*3.4. Analysis*

The analysis for the current paper includes the descriptive statistics of all dependent and independent variables for the MSAs examined here, followed by breakdowns of descriptive statistics for sanctuary versus non-sanctuary status MSAs. We then present the multivariate analysis, which analyzes the relationship between immigration levels, sanctuary status, and overdose deaths overall and for Latinx, Black, and White net controls. Due to the rare nature of overdose deaths and positive skewness in these measures, our multivariate analysis relies on negative binomial regression. This decision was affirmed in analysis of the dependent variables, as overdispersion was present (variance exceeds the mean for all 4 dependent variables). The analysis of VIF scores were at or below the standard cutoff levels (below 4.0), indicating that collinearity was not a concern.

## 4. Results

*4.1. Descriptive Statistics*

Descriptive statistics are shown in Table 1 for the overall sample as well as for sanctuary MSAs and non-sanctuary MSAs. Focusing first on the overdoses for the sample as a whole, MSAs had an average of 153.98 total overdose deaths, which translated to a rate of 18.16 per 100,000. However, we find that overdose deaths varied considerably between racial and ethnic groups for our sample of MSAs. Latinx overdose deaths averaged 13.38, with a rate of 7.80 per 100,000, non-Latinx Black deaths averaged 17.7 with a rate of 16.63 per 100,000, and non-Latinx Whites averaged 118.36 deaths with a rate of 21.80 per 100,000. Notably, this aligns with prior research, indicating that White populations have seen particularly high levels of overdose mortality [93–96].

Regarding independent variables of interest, MSA populations had 4.47% Latinx foreign-born residents on average, ranging from 0.12% (Wheeling-WV-OH MSA) to a maximum of 44.13% (Miami, Florida MSA). Of the 292 MSAs examined here, 13.7% had some form of formal, written sanctuary policy in place.

Table 1 also reveals some key differences in our variables of interest across sanctuary and non-sanctuary MSAs. In general, overdose deaths were more common in sanctuary jurisdictions, overall and across most race/ethnic groups. Specifically, Table 1 shows that sanctuary MSAs had an average overdose death rate of 19.54 per 100,000, compared to non-sanctuary MSAs that had an average of 17.94 overdose deaths per 100,000. Similarly, sanctuary jurisdictions had higher Black and Latinx overdose rates, but White overdose death rates were similar across sanctuary and non-sanctuary MSAs. With respect to our predictors, sanctuary MSAs had higher levels of Latinx immigration. They also had much larger populations, which may explain the higher overdose rates described above (as they tend to be more urban). Sanctuary MSAs had a greater share of uninsured persons. Sanctuary jurisdictions also had lower levels of White disadvantage and opioid prescription rates, but do not differ for other variables.

**Table 1.** Summary Statistics.

| | Total | | Sanctuary MSA | | Non-Sanctuary MSA | |
|---|---|---|---|---|---|---|
| | Mean (Rate) | SD | Mean (Rate) | SD | Mean (Rate) | SD |
| **Dependent Variables** | | | | | | |
| Total overdose deaths | 153.98 (18.16) | 153.98 | 454.23 (19.54) | 451.51 | 106.33 (17.94) | 135.78 |
| Latinx overdose deaths | 13.38 (7.80) | 32.31 | 45.43 (11.79) | 65.04 | 8.28 (7.17) | 19.05 |
| White overdose deaths | 118.36 (21.80) | 177.40 | 322.25 (22.5) | 329.55 | 85.99 (21.69) | 109.25 |
| Black overdose deaths | 17.70 (16.63) | 43.01 | 70.26 (26.09) | 94.11 | 9.36 (15.12) | 16.09 |
| **Independent Variables** | | | | | | |
| Latinx foreign-born | 4.47% | 5.56 | 5.99% | 4.90 | 4.23% | 5.63 |
| Sanctuary status | 13.70% | | | | | |
| Total population | 878,351 | 1,399,192 | 2,607,703 | 2,655,185 | 603,850.80 | 788,708 |
| Opioid prescription rate | 51.76 | 45.04 | 20.18 | 17.40 | 56.78 | 46.05 |
| Total uninsured | 79,656.17 | 141,910.20 | 199,533.50 | 266,897.80 | 60,627.95 | 98,111.44 |
| Residential mobility | 0.16 | 3.67 | 14.57 | 3.14 | 16.17 | 3.71 |
| Police presence per 100,000 | 233.77 | 156.77 | 257.34 | 171.35 | 230.02 | 154.37 |
| *Disadvantage Index-Total* | 11.58 | 2.80 | 11.72 | 2.80 | 10.73 | 2.68 |
| Unemployed | 6.54% | 1.72 | 6.55% | 1.77 | 6.49% | 1.33 |
| Female headed household | 12.64% | 2.79 | 12.79% | 2.83 | 11.69% | 2.26 |
| Population in poverty | 15.21% | 4.07 | 15.47% | 3.99 | 13.59% | 4.20 |
| No high school diploma | 11.92% | 5.02 | 12.05% | 5.11 | 11.14% | 4.46 |
| *Disadvantage Index-Latinx* | 20.50 | 4.17 | 19.71 | 3.62 | 20.63 | 4.24 |
| Unemployed | 7.80% | 2.92 | 7.95% | 2.00 | 7.79% | 3.05 |
| Female headed household | 19.42% | 5.98 | 19.78% | 5.18 | 19.36% | 6.10 |
| Population in poverty | 24.45% | 6.60 | 21.72% | 5.50 | 24.88% | 6.67 |
| No high school diploma | 30.34% | 9.99 | 29.39% | 7.88 | 30.50% | 10.29 |
| *Disadvantage Index-White* | 8.03 | 1.62 | 6.76 | 1.51 | 8.23 | 1.53 |
| Unemployed | 5.37% | 1.28 | 5.23% | 1.05 | 5.39% | 1.31 |
| Female headed household | 8.45% | 1.62 | 7.36% | 1.48 | 8.63% | 1.58 |
| Population in poverty | 10.82% | 3.00 | 9.04% | 3.11 | 11.11% | 2.89 |
| No high school diploma | 7.46% | 2.78 | 5.42% | 1.82 | 7.78% | 2.77 |
| *Disadvantage Index-Black* | 21.40 | 5.51 | 19.64 | 5.02 | 21.67 | 5.54 |
| Unemployed | 11.83% | 4.11 | 11.87% | 3.20 | 11.82% | 4.25 |
| Female headed household | 30.31% | 9.69 | 28.88% | 9.60 | 30.54% | 9.70 |
| Population in poverty | 28.38% | 8.62 | 25.04% | 7.41 | 28.91% | 8.69 |
| No high school diploma | 15.06% | 5.27 | 12.79% | 3.79 | 15.42% | 5.39 |
| | N = 292 | | n = 40 | | n = 252 | |

*Notes*: Dependent variables are calculated based on 3-year averages and presented in counts with rate/100,000 persons in parentheses.

### 4.2. Multivariate Analysis

Turning to our multivariate analysis, Table 2 displays negative binomial regression results for total overdose deaths, as well as for Latinx, Black, and White overdose deaths. For each group, Model 1 addresses our first hypothesis about immigration effects on overdose and includes immigrant concentration with all controls. Model 2 adds the measure of sanctuary city status to address our second hypothesis. Last, Model 3 includes an interaction term between immigrant concentration and sanctuary status to address our third hypothesis and assess whether relationships between immigrant concentration and overdose depend on sanctuary status.

Turning first to Model 1 and focusing on our key variable of interest, we find that Latinx immigrant concentration is generally associated with lower overdose rates overall and across race/ethnicity. In our "total" population models, Table 2 shows that immigrant concentration is associated with lower levels of overdose deaths. For each additional 1% increase in the foreign-born Latinx population, total overdose deaths are reduced by approximately 3% (IRR = 0.972, $p < 0.001$). Similar effects are shown for both Black and White overdoses as well. That is, MSAs with higher concentrations of Latinx immigrants had significantly lower levels of Black and White overdoses deaths, net of other factors. The one exception is for Latinx overdose deaths, which were significantly higher in areas with larger Latinx foreign-born populations. Substantively, these effects indicate that, for each 1 percent increase in the Latinx foreign born within an MSA, Black and White overdose deaths are 4 to 5 percent lower (respectively) but Latinx overdose deaths are approximately 10 percent higher (IRR = 1.10, $p < 0.001$).[7] Thus, we generally find that immigration is

linked to lower overdose deaths (as suggested by the immigrant revitalization perspective), except for Latinx overdose deaths.

With respect to the control variables, Table 2 shows several additional noteworthy effects. The total population was associated with higher overdose death rates overall and for all race/ethnic groups examined here. The opioid prescription rate was linked to lower overdose mortality for Latinx and White populations. Residential mobility and total uninsured were associated with lower overdose deaths overall and for White populations. Other predictors had different effects across race/ethnicity. The disadvantage index was associated with higher levels of White and Black overdose deaths. Nearly all other controls were non-significant in the analysis of Latinx overdose deaths (disadvantage, police presence, residential mobility, and uninsured population).

Regarding the second hypothesis, Model 2 in Table 2 shows that sanctuary status generally was not significantly associated with overdose deaths across most models. Sanctuary status had no significant effect on overdose deaths for the "total", Black, and White models. In contrast, Table 2 indicates that Latinx overdose death rates were significantly higher in MSAs with sanctuary status. Specifically, Latinx overdose deaths were approximately two times greater in MSAs with sanctuary status compared to non-sanctuary MSAs (IRR = 1.997, $p < 0.001$). Notably, these findings conflict with expectations drawn from both the immigrant revitalization perspective and systems avoidance theory, which suggested that sanctuary jurisdictions may be more protective for overdose mortality (an issue we return to in our discussion).

Last, we turn to Model 3 in Table 2, which includes an interaction term between Latinx immigrant concentration and sanctuary status. This assesses our third hypothesis and explores whether the relationship between immigrant concentration and overdose mortality depends on the sanctuary status of the MSA. Model 3 for each group shows that this interaction term is not significant, indicating that the ties between percent Latinx foreign born and overdose deaths do not differ in sanctuary versus non-sanctuary MSAs. We also note here that the main effect of sanctuary status drops out of significance for Latinx overdose deaths. Aside from this change, the remaining variables were largely unchanged from Models 1 and 2 across groups with some slight changes to sizes of coefficients.

**Table 2.** Negative Binomial Regression of Expected Counts of Total, Latinx, Black, and White Overdose Deaths.

| | Total | | | | | | Latinx | | | | | | Black | | | | | | White | | | | | |
|---|---|---|---|---|---|---|---|---|---|---|---|---|---|---|---|---|---|---|---|---|---|---|---|---|
| | Model 1 | | Model 2 | | Model 3 | | Model 1 | | Model 2 | | Model 3 | | Model 1 | | Model 2 | | Model 3 | | Model 1 | | Model 2 | | Model 3 | |
| | b (SE) | IRR | b (SE) | IRR | b (SE) | IRR | b (SE) | IRR | b (SE) | IRR | b (SE) | IRR | b (SE) | IRR | b (SE) | IRR | b (SE) | IRR | b (SE) | IRR | b (SE) | IRR | b (SE) | IRR |
| % Latinx foreign-born | −0.028 *** (0.005) | 0.972 | −0.028 *** (0.005) | 0.972 | −0.028 *** (0.006) | 0.972 | 0.096 *** (0.014) | 1.100 | 0.093 *** (0.013) | 1.098 | 0.091 *** (0.014) | 1.095 | −0.037 *** (0.008) | 0.964 | −0.036 *** (0.009) | 0.964 | −0.036 *** (0.009) | 0.965 | −0.049 *** (0.005) | 0.953 | −0.049 *** (0.005) | 0.952 | −0.048 *** (0.005) | 0.953 |
| Sanctuary status | | | 0.006 (0.071) | 1.006 | 0.062 (0.107) | 1.064 | | | 0.692 *** (0.161) | 1.997 | 0.528 (0.281) | 1.696 | | | 0.042 (0.121) | 1.043 | 0.120 (0.175) | 1.128 | | | −0.053 (0.074) | 0.948 | 0.027 (0.111) | 1.028 |
| Foreign born x Sanctuary status | | | | | −0.010 (0.014) | 0.990 | | | | | 0.027 | 1.027 | | | | | −0.015 | 0.985 | | | | | −0.015 | 0.985 |
| Total population [a] | 1.165 *** (0.037) | 3.205 | 1.164 *** (0.038) | 3.203 | 1.154 *** (0.040) | 3.172 | 0.915 *** (0.105) | 2.498 | 0.837 *** (0.106) | 2.309 | 0.862 *** (0.113) | 2.368 | 1.453 *** (0.069) | 4.276 | 1.446 *** (0.071) | 4.247 | 1.430 *** (0.075) | 4.181 | 1.184 *** (0.038) | 3.267 | 1.190 *** (0.039) | 3.286 | 1.176 *** (0.041) | 3.241 |
| Opioid prescription rate | 2.28E-4 (0.001) | 1.000 | 2.418E-4 (0.001) | 1.000 | 2.109E-4 (0.001) | 1.000 | −0.010 *** (0.003) | 0.990 | −0.010 *** (0.002) | 0.990 | −0.010 *** (0.002) | 0.990 | 0.002 (0.001) | 1.002 | 0.002 (0.001) | 1.002 | 0.002 (0.001) | 1.002 | −0.002 * (0.001) | 0.998 | −0.001 * (0.001) | 0.998 | −0.002 * (0.001) | 0.998 |
| Residential mobility | −0.019 ** (0.007) | 0.981 | −0.019 ** (0.007) | 0.982 | −0.019 ** (0.007) | 0.981 | −0.017 (0.018) | 0.983 | 0.002 (0.018) | 1.002 | 0.004 (0.018) | 1.004 | −0.018 (0.013) | 0.982 | −0.017 (0.013) | 0.983 | −0.018 (0.013) | 0.982 | −0.020 ** (0.007) | 0.980 | −0.021 ** (0.007) | 0.979 | −0.021 ** (0.007) | 0.979 |
| Disadvantage index [b] | 0.006 (0.011) | 1.006 | 0.006 (0.011) | 1.006 | 0.007 (0.011) | 1.007 | −0.007 (0.016) | 0.994 | 4.679E-4 (0.015) | 1.000 | 0.001 (0.015) | 1.001 | 0.053 *** (0.010) | 1.054 | 0.053 *** (0.010) | 1.054 | 0.052 *** (0.010) | 1.054 | 0.112 *** (0.019) | 1.118 | 0.110 *** (0.019) | 1.117 | 0.110 *** (0.019) | 1.117 |
| Total uninsured | −9.260E-7 *** (0.000) | 1.000 | −9.270E-7 *** (0.000) | 1.000 | −8.390E-7 ** (0.000) | 1.000 | −3.600E-7 (0.000) | 1.000 | −2.69E-7 (0.000) | 1.000 | −4.440E-7 (0.000) | 1.000 | −3.870E07 (0.000) | 1.000 | −3.930E-7 (0.000) | 1.000 | −2.570E-7 (0.000) | 1.000 | −9.720E-7 *** (0.000) | 1.000 | −9.650E-7 *** (0.000) | 1.000 | −8.340E-7 ** (0.000) | 1.000 |
| Police presence | −2.456E-4 (0.000) | 1.000 | −2.456 (0.000) | 1.000 | −2.454E-4 (0.000) | 1.000 | 1.405E-4 (0.000) | 1.000 | −1.649E-4 (0.000) | 1.000 | −1.734E-7 (0.000) | 1.000 | 0.001 *** (0.000) | 1.001 | 0.001 *** (0.000) | 1.001 | 0.001 *** (0.000) | 1.001 | −2.95E-4 (0.000) | 1.000 | −2.905E-4 (0.000) | 1.000 | −2.887E-4 (0.000) | 1.000 |
| Constant | −10.333 *** (0.544) | 0.000 | −10.327 *** (0.548) | 0.000 | −10.210 *** (0.571) | 0.000 | −10.008 *** (1.624) | 0.000 | −9.483 (1.603) | 0.000 | −9.824 (1.680) | 0.000 | −18.222 *** (1.084) | 0.000 | −18.15278 *** (1.100) | 0.000 | −17.929 *** (1.154) | 0.000 | −11.4042 *** (0.572) | 0.000 | −11.450 *** (0.574) | 0.000 | −11.274 *** (0.598) | 0.000 |
| Pseudo $R^2$ | 0.208 *** | | 0.208 *** | | 0.209 *** | | 0.201 *** | | 0.211 *** | | 0.211 *** | | 0.259 *** | | 0.259 *** | | 0.259 *** | | 0.206 *** | | 0.206 *** | | 0.206 *** | |

N = 292. *Notes*: * $p < 0.05$, ** $p < 0.01$, *** $p < 0.001$. [a] Denotes a log transformation. [b] Denotes a race-specific index.

### 5. Discussion

The current paper contributes to a growing body of knowledge on how immigration may impact serious drug use and overdose mortality [37] both overall and across different racial and ethnic groups (specifically Latinx, Black, and White populations). In addition, we extend this line of research by assessing the influence of sanctuary status in immigration–overdose relationships and incorporating systems avoidance theory as a framework for understanding potential effects of immigration and sanctuary status on overdose risk. Our analysis revealed several noteworthy findings.

In line with the immigrant revitalization perspective, total overdose deaths were significantly lower in MSAs with higher Latinx immigrant concentrations. Similarly, Black and White overdoses were also lower in areas with larger Latinx foreign-born populations, and these effects remained even after controlling for other relevant contextual factors such as disadvantage. However, Latinx overdose mortality was actually higher in places with larger shares of immigrants, which we return to below. In contrast to our predictions, sanctuary status had little association with overdose mortality across most of our models. The one exception was for Latinx overdoses, where sanctuary status was surprisingly linked to higher overdoses. Last, interactions between sanctuary status and immigration were all non-significant, indicating that the relationships between immigrant concentration and overdose do not differ markedly between sanctuary and non-sanctuary MSAs.

These findings provide several noteworthy implications for research, theory, and broader concerns about the overdose crisis. First, the results of our study conflict with narratives and political rhetoric suggesting that immigration is a driver of the overdose crisis. Despite public fears and political commentary linking immigration and drug problems, our findings support an emerging body of research indicating that immigrant concentration is not linked to greater overdose deaths [11,37]. For all but one group, Latinx immigration was actually associated with lower levels of overdose death (overall and for Black and White populations). The one exception to this pattern was for Latinx overdose deaths. However, outside of this effect, we find no indication that immigration is linked to greater overdose mortality, and most of the findings suggest the opposite. Overdose deaths have been driven by a variety of social and economic factors, but the current findings suggest that immigration has not been one of them. As such, policies aimed at broad-based immigration control as a tool for battling the overdose crisis seem ill-suited for reducing overdose mortality.

Second, the findings presented here provide support for immigrant revitalization perspectives and suggest that the protective effects of immigration may extend beyond traditional measures of crime to also affect drug overdose deaths. A growing body of research has found that immigrant concentrations provide an insulating effect against crime and related social problems (including public health outcomes [11,37]). However, few analyses to date have assessed whether these same insulating effects apply to the overdose crisis. Our findings suggest that they do. Thus, there is growing indication that immigrant revitalization has broad based effects that protect communities from a wide range of social ills, including serious drug related deaths.

Third, this analysis highlights the importance of considering systems avoidance theory, though the effects are more complicated and mixed here. The increased number of Latinx overdose deaths in places with larger foreign-born populations is consistent with the systems avoidance framework. Systems avoidance theory suggests that help-seeking behavior from record-keeping institutions may be suppressed by fears of detection and deportation [14], *perhaps more so for Latinx populations* who have larger shares of immigrants and greater risk of deportation. Thus, the fact that immigrant concentration was associated with higher Latinx overdoses is precisely what systems avoidance theory would predict. For highly-immigrant groups like Latinx populations, systems avoidance could contribute to elevated overdose risk due to fears about contacting official agencies.

However, the effects of sanctuary status observed here were somewhat unexpected and do not align with predictions from systems avoidance theory. Systems avoidance theory (and immigrant revitalization) suggests that sanctuary cities may help reduce overdose risk and bolster the protective effects of immigration on overdoses. Yet, sanctuary status generally had little impact on overdose mortality or on immigration–overdose relationships in our models, and it was actually linked to higher Latinx overdoses, in contrast to our predictions. Although our analysis suggests that there is value in systems avoidance theory for studies of immigration, crime, and drug use, much more work is needed that explicitly tests and assesses systems avoidance effects.

Last, the findings for Latinx overdose merit some further discussion. In contrast to our models of total, Black, and White overdose, immigrant concentration and sanctuary status were linked to higher levels of Latinx overdose. This finding could be interpreted in several ways. On the one hand, it could mean that immigration and sanctuary status contribute to higher levels of Latinx overdose. As previously discussed, Latinx immigrant populations are at the greatest risk of deportation in the U.S. [54] and thus may be less likely to seek help for addiction from formal institutions as would be predicted by systems avoidance. Even among those who are fully documented, there are reasons to avoid detection from authorities. For example, Latinx populations in particular may be hesitant to bring attention to family or other community members who may not themselves be fully documented. Furthermore, from a Latinx critical criminology perspective, Vélez and Peguero [97] point out there is an overall perception among some that Latinx communities have been criminalized, which discourages trust in formal institutions. In particular, Latinx "enclaves" tend to undergo increased surveillance and experience more frequent ICE raids, despite overall lower crime rates in these communities [98]. Although criminal justice institutions have been a primary source of this tension, it is possible that it could extend to hospitals and other official sources of substance abuse treatment. Initially, we hypothesized that sanctuary status may reduce distrust and encourage help-seeking. However, our findings suggest that benefits to help-seeking behavior found in prior literature regarding victimization reporting [49,78] do not necessarily extend to overdose mortality. There are a few possible reasons why this might be the case, on which we can only speculate. It could be, for example, that sanctuary policies are not viewed as protective of immigrants in the medical setting, particularly when situations involve drug use, since deportation can occur for mere drug possession [99]. In fact, some research suggests that even informal sources of help may be deemed too risky given the precarious status of those with both addiction and uncertain documentation status [100]. While the opioid crisis has brought calls for compassionate treatment for those with addiction issues [13], this compassion may not apply to undocumented immigrants—which could ultimately result in higher numbers of overdose deaths for Latinx populations specifically in sanctuary jurisdictions.

On the other hand, our results may simply reflect a selection effect. MSAs with larger shares of immigrants and sanctuary policies may seem more welcoming to immigrants and especially to more vulnerable immigrant populations. Thus, rather than causing higher Latinx overdose deaths, these findings may indicate that Latinx populations who are more at risk for overdose simply tend to settle in places with large immigrant populations and sanctuary policies. It is also possible that increased Latinx overdose deaths in sanctuary areas are unrelated to help-seeking behavior, and there may be additional contextual factors that were not accounted for in the model due to data limitations. One such possibility is that these areas may have fewer community health resources available to them. The current study is unable to assess these different possibilities, but it is an important avenue for study in future research and especially in studies that examine these relationships using longitudinal data.

*Limitations and Directions for Future Research*

Although the current study makes several contributions to research on immigration, crime, and macro-level patterns of overdose, it is not without limitations. First, this study extends prior research by incorporating systems avoidance theory, but as noted above, we are unable to directly test the mechanisms outlined by this framework. Systems avoidance theory informs our analysis and describes potential effects of immigration and sanctuary status on overdose risks. However, more work is needed that directly assesses the ways in which immigration and sanctuary status impact help seeking behavior for drug use, and in turn, overdose risk.

Second, the cross-sectional nature of the analysis limits our ability to identify the causal direction of effects identified here. Rather than increasing or decreasing overdoses, it may be that immigrant communities and sanctuary cities tend to attract populations that are more or less at risk of overdose. As noted above, this seems particularly plausible for Latinx populations, where the more vulnerable groups (undocumented populations, disadvantaged, lower income, fewer social ties) may gravitate to places that seem more welcoming. Future research that uses longitudinal analyses and examines changes in overdose rates following changes in immigration and (before/after) the adoption of sanctuary policies is needed to better establish the causal ordering of effects and the mechanisms that may be at work in these relationships.

Although it was beyond the scope of the current paper, there are several other variables and characteristics of places that should be examined in future analysis of immigration, sanctuary status, and drug use. Future research should examine these relationships for undocumented versus documented immigrant populations. Drawing on systems avoidance theory, we might expect these effects and patterns of systems avoidance to differ between documented and undocumented groups. Research should also explore these relationships across traditional versus new immigrant destinations. Xie and Baumer [8] found that there were distinct differences in crime reporting behavior between traditional destinations (areas with a history of large numbers of immigrants) and new destinations. Thus, there could be parallels in other forms of help-seeking behavior that may be related to overdose mortality. Last, alternative measures of sanctuary status (including informal sanctuary policies) should be considered in future work to further assess the impact of sanctuary policies on overdose or related behaviors.[8]

## 6. Conclusions

Clearly, much more work is needed to develop a comprehensive understanding of the macro-level relationships between immigration, sanctuary policies, and overdose mortality. The current study offers an important step toward this goal and suggests that, in line with immigrant revitalization perspectives, immigration generally is not linked to greater overdose rates for most groups examined here. Further exploration is needed to better understand whether and how sanctuary policies impact help seeking behavior and overdose risk. As suggested here, systems avoidance theory may be a useful framework for exploring these relationships. Although for now, both immigration and sanctuary status do not appear to be consistent drivers of overdose mortality as a whole.

**Author Contributions:** Conceptualization: K.P.; Methodology: D.S., B.F. and K.P.; Analysis: K.P. and D.S.; Data curation: D.S. and K.P.; Writing: original draft preparation: K.P., Review/edits/rewrite: B.F., D.S. and K.P.; Visualization: D.S.; Supervision: B.F. All authors have read and agreed to the published version of the manuscript.

**Funding:** This material is based upon work supported by the National Science Foundation under Grant No. 1849209. Any opinions, findings, and conclusions or recommendations expressed in this material are those of the authors and do not necessarily reflect the views of the National Science Foundation.

**Institutional Review Board Statement:** Not applicable.

**Informed Consent Statement:** Not applicable.

**Data Availability Statement:** 3rd Party Data: Restrictions apply to the availability of these data. Data was obtained in part from CDC restricted access data and are not publicly available.

**Conflicts of Interest:** The authors declare no conflict of interest.

## Notes

1. Overdose death data were calculated based on an individual's county of residence. These data were then aggregated to the MSA-level using a crosswalk file. Overdose death counts were based on underlying death codes following CDC categorizations and prior research [89,90]: overdose, X40 to X44, X60 to X64, X85, and Y10 to Y14.
2. Coefficients were slightly larger using % Latinx foreign born than overall % foreign born for all groups, but significance levels were unchanged.
3. There is no standard definition of a sanctuary area. We use a conservative approach and only count those areas with formalized sanctuary resolutions in place as having sanctuary status. However, we conducted additional analyses using a sanctuary variable that included all MSAs that reported non-cooperation with ICE in 2015 that could be located (even without formal resolutions). Results did not change substantively.
4. MSAs containing large sanctuary cities, such as San Francisco, were almost all coded as having sanctuary status while mixed status areas where it was unclear how much of the MSA was covered by sanctuary policies were excluded. Most areas with sanctuary policies were coded as sanctuary MSAs due to the tendency for these policies to be enacted in large cities or counties (see [16]). However, for 10 MSAs, coverage of sanctuary policies were unclear or the sanctuary policy in question did not cover at least half the total population included in the MSA.
5. For information on sanctuary data sources, see FAIR US https://www.fairus.org/sites/default/files/2017-08/Sanctuary_Policies_Across_America_Report.pdf (accessed on 18 March 2023), Center for Immigration Studies https://cis.org/Map-Sanctuary-Cities-Counties-and-States (accessed on 18 March 2023, and Ohio Jobs and Justice PAC http://ojjpac.org/sanctuary.asp (accessed on 18 March 2023).
6. Indices were created with the "alpha" command in Stata, and were not standardized given the consistent measurement between indicators. Alternate indices (standardized alpha and factors using principal components analysis) were also examined in supplemental analyses. Results were consistent regardless of how the indices were created, though the effects of disadvantage had slightly stronger effects when using principal components analysis.
7. Because there was a concern that this may be a function of fewer White residents naturally resulting from a larger % of Latinx foreign-born residents, supplemental analysis was also run controlling for total non-Latinx White population rather than total population, which revealed that controlling for this only made the *% Latinx foreign-born* effect stronger (supplemental analysis not shown, available upon request).
8. A second, less stringently defined sanctuary variable was created for supplemental analysis in which MSAs with any sanctuary policy (formal or informal) and declined detainer requests as reported by Fair U.S. and the Center for Immigration Studies were coded as sanctuaries. Results from the supplemental models did not change substantively.

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
