# Peer review of "Staying under the Radar? Immigration Effects on Overdose Deaths and the Impact of Sanctuary Jurisdictions"

_societies, doi:10.3390/soc13060135_

Round 1

Reviewer 1 Report

The author(s) set out to examine immigration concentration, sanctuary status, and overdose mortality across MSA. The dataset is quite interesting, and methods are sound. Good use of multivariate and bivariate analyses. This paper is thorough, and will bring a needed contribution to a large gap in the literature. My only confusion that I may have missed, is that the authors talk about MSAs, but based on what I rad never directly state what an MSA is. Hopefully I just missed this, but if not, it needs to be clearly defined. 

Author Response

1. “My only confusion that I may have missed, is that the authors talk about MSAs, but based on what I read never directly state what an MSA is. Hopefully I just missed this, but if not, it needs to be clearly defined.”  

RESPONSE: We thank Reviewer 1 for bringing this to our attention. To clarify what an MSA is, we added a full explanation of what an MSA entails on page 8 under the materials and methods section. We also include a reference to other research of this nature at the MSA level for the audience.

Thanks for taking the time to review our paper!

Reviewer 2 Report

This is a well-written, well-researched article on a timely topic of great political importance. The research question is original and compelling and the findings provocative. I have only a couple of comments.

The first is that the authors give Trump's anti-immigrant comments a lot of play in this article, suggesting a strong connection between anti-immigrant politics and sanctuary cities. Yet the data set they use pre-dates Trump. While there is some mention of the long-standing history of anti-immigrant policies in the United States, I think it would be worth laying the history of sanctuary policies in a bit more depth. It might also be worth noting that fear of deportation predated Trump, and was quite high under Obama who was in office when the data set was constructed.

My second comment is about the finding that only Latinx overdose deaths went up in communities with high immigrant populations. This is puzzling, and I am not satisfied with the few sentences of analysis the authors give to this finding. Sure, maybe this is because Latinos who are more vulnerable are more attracted to live in sanctuary cities. But what are the other possibilities. I think the authors need to think (and read) more about this, in order to offer the reader a more substantive discussion of this finding. 

Author Response

  1. “The authors give Trump's anti-immigrant comments a lot of play in this article, suggesting a strong connection between anti-immigrant politics and sanctuary cities. Yet the data set they use pre-dates Trump. While there is some mention of the long-standing history of anti-immigrant policies in the United States, I think it would be worth laying the history of sanctuary policies in a bit more depth. It might also be worth noting that fear of deportation predated Trump, and was quite high under Obama who was in office when the data set was constructed.”

RESPONSE 1: This is a good point, and we have expanded our discussion on the history of anti-immigrant policies in the United States on page 6, section 2.4 Sanctuary Jurisdictions.

“Though political rhetoric has brought renewed attention to sanctuary jurisdictions [O’brien et. al, 2019; Roy, 2019], the initial sanctuary movement in the U.S. occurred in the 1980s, and was largely driven by religious institutions offering aid and protection from immigration authorities to undocumented refugees fleeing El Salvador and Guatemala [Paik, 2017]. In 2007, renewed interest in the sanctuary movement occurred [Paik, 2017] as the criminalization of migrants increased, and the use of incarceration without due process expanded. Additionally, there was a rapid expansion of border patrol [Goodman, 2020], met with heightened concerns about protecting immigrants. This concern continued throughout the 2000s and 2010s as the country ramped up aggressive deportation and the removal of immigrants throughout the Obama administration. More recently, there was a large spike in enacted sanctuary policies in response to Trump’s declared anti-immigration stance [Rose, 2021].”

Additionally, we discuss how policies have changed over time in more detail between 2009 and 2017 and provide some examples.

  1. My second comment is about the finding that only Latinx overdose deaths went up in communities with high immigrant populations. This is puzzling, and I am not satisfied with the few sentences of analysis the authors give to this finding. Sure, maybe this is because Latinos who are more vulnerable are more attracted to live in sanctuary cities. But what are the other possibilities. I think the authors need to think (and read) more about this, in order to offer the reader a more substantive discussion of this finding.

RESPONSE 2: We thank the reviewer for this suggestion. Although we recognize the limitations to our data, we provide a much more in-depth discussion of these findings in which we propose a few additional reasons for why we might find higher levels of Latinx overdose deaths both in communities with higher immigrant populations and in sanctuary jurisdictions based on extant literature.

Thanks for taking the time to review our paper!

Reviewer 3 Report

This is a well written, researched and thouroughly cited paper.

Two things to consider:

1) There is some rhetoric that needs evidence. The authors are citing plenty in most sections of the article, yet make some sweeping statements without any backup e.g. "Immigrant populations tend to be disadvantaged and have greater educational and financial deprivation than many native-born populations. They may also face a variety of non-financial strains and challenges associated with settlement in a new environment (e.g., language barriers, navigating immigration processes, finding work, housing, transportation, and childcare). In addition, the influx of disadvantaged groups with different languages and norms could inhibit community organization and informal control over crime according to traditional social disorganization arguments." - that is a very loaded paragraph and seems to need lots of sources if not to sound assumptive/stereptyping.

2) Authors could take a look at international research that suggests that crime can be lower in areas with high concentration of immigrants, e.g.

Ignatans, D., & Roebuck, T. (2018). Do more immigrants equal more crime?: Drawing a bridge between first generation immigrant concentration and recorded crime rates. Crime Security and Society, 1(1), 22-40.

Of course these rates are low due to self policing that takes place in those areas, but it is a point worth considering.

Author Response

Point 1: There is some rhetoric that needs evidence. The authors are citing plenty in most sections of the article, yet make some sweeping statements without any backup e.g. "Immigrant populations tend to be disadvantaged and have greater educational and financial deprivation than many native-born populations. They may also face a variety of non-financial strains and challenges associated with settlement in a new environment (e.g., language barriers, navigating immigration processes, finding work, housing, transportation, and childcare). In addition, the influx of disadvantaged groups with different languages and norms could inhibit community organization and informal control over crime according to traditional social disorganization arguments." - that is a very loaded paragraph and seems to need lots of sources if not to sound assumptive/stereotyping.

RESPONSE 1: We thank the reviewer for bringing this to our attention. We have included several citations to the specified paragraph located on page 4, Section 2.2 Reasons Immigration Could Increase Overdose Deaths to support these claims.

Point 2: Authors could take a look at international research that suggests that crime can be lower in areas with high concentration of immigrants, e.g.  Ignatans, D., & Roebuck, T. (2018). Do more immigrants equal more crime?: Drawing a bridge between first generation immigrant concentration and recorded crime rates. Crime Security and Society, 1(1), 22-40. Of course these rates are low due to self policing that takes place in those areas, but it is a point worth considering.

RESPONSE 2: We thank the reviewer for the Ignatans & Roebuck (2018) article recommendation and have incorporated the article on page 6, section 2.3 Reasons Immigration Could Decrease Overdose Deaths. Our study focuses primarily on the U.S. context due to the nature of our data/theoretical framework; however, we included some citations that refer to additional international literature as requested to address this shortcoming.

Thanks for taking the time to review our paper!

Round 2

Reviewer 2 Report

This is an excellent revision.